# Bone Marrow Microenvironment in Light-Chain Amyloidosis: In Vitro Expansion and Characterization of Mesenchymal Stromal Cells

**DOI:** 10.3390/biomedicines9111523

**Published:** 2021-10-22

**Authors:** Chiara Valsecchi, Stefania Croce, Alice Maltese, Lorenza Montagna, Elisa Lenta, Alice Nevone, Maria Girelli, Paolo Milani, Tiziana Bosoni, Margherita Massa, Carlotta Abbà, Rita Campanelli, Jessica Ripepi, Annalisa De Silvestri, Adriana Carolei, Giovanni Palladini, Marco Zecca, Mario Nuvolone, Maria Antonietta Avanzini

**Affiliations:** 1Pediatric Hematology Oncology, Cell Factory, Fondazione IRCCS Policlinico S. Matteo, 27100 Pavia, Italy; c.valsecchi@smatteo.pv.it (C.V.); a.maltese@smatteo.pv.it (A.M.); e.lenta@smatteo.pv.it (E.L.); m.zecca@smatteo.pv.it (M.Z.); 2General Surgery Department, Fondazione IRCCS Policlinico S. Matteo, 27100 Pavia, Italy; s.croce@smatteo.pv.it; 3Department of Clinical, Surgical, Diagnostic & Pediatric Sciences, University of Pavia, 27100 Pavia, Italy; lorenza.montagna@unpv.it; 4General Medicine 2—Center for Systemic Amyloidoses and High-Complexity Diseases, Fondazione IRCCS Policlinico S. Matteo, 27100 Pavia, Italy; alice.nevone01@universitadipavia.it (A.N.); maria.girelli01@universitadipavia.it (M.G.); p.milani@smatteo.pv.it (P.M.); m.massa@smatteo.pv.it (M.M.); carlotta.abba@outlook.it (C.A.); r.campanelli@smatteo.pv.it (R.C.); jessica.ripepi01@universitadipavia.it (J.R.); a.carolei@smatteo.pv.it (A.C.); giovanni.palladini@unipv.it (G.P.); mario.nuvolone@unipv.it (M.N.); 5Department of Molecular Medicine, University of Pavia, 27100 Pavia, Italy; 6Clinical Chemistry Laboratory, Fondazione IRCCS Policlinico S. Matteo, 27100 Pavia, Italy; t.bosoni@smatteo.pv.it; 7Clinical Epidemiology and Biometry Unit, Fondazione IRCCS Policlinico S. Matteo, 27100 Pavia, Italy; a.desilvestri@smatteo.pv.it

**Keywords:** light-chain amyloidosis, mesenchymal stromal cells, bone marrow microenvironment, immunomodulation

## Abstract

Immunoglobulin light-chain amyloidosis (AL) is caused by misfolded light chains produced by a small B cell clone. Mesenchymal stromal cells (MSCs) have been reported to affect plasma cell behavior. We aimed to characterize bone marrow (BM)-MSCs from AL patients, considering functional aspects, such as proliferation, differentiation, and immunomodulatory capacities. MSCs were in vitro expanded from the BM of 57 AL patients and 14 healthy donors (HDs). MSC surface markers were analyzed by flow cytometry, osteogenic and adipogenic differentiation capacities were in vitro evaluated, and co-culture experiments were performed in order to investigate MSC immunomodulatory properties towards the ALMC-2 cell line and HD peripheral blood mononuclear cells (PBMCs). AL-MSCs were comparable to HD-MSCs for morphology, immune-phenotype, and differentiation capacities. AL-MSCs showed a reduced proliferation rate, entering senescence at earlier passages than HD-MSCs. The AL-MSC modulatory effect on the plasma-cell line or circulating plasma cells was comparable to that of HD-MSCs. To our knowledge, this is the first study providing a comprehensive characterization of AL-MSCs. It remains to be defined if the observed abnormalities are the consequence of or are involved in the disease pathogenesis. BM microenvironment components in AL may represent the targets for the prevention/treatment of the disease in personalized therapies.

## 1. Introduction

The term “systemic amyloidosis” comprehends a number of diseases caused by the extracellular deposition of misfolded proteins in vital organs, resulting in their dysfunction [1].

Among them, immunoglobulin light-chain amyloidosis (AL), the most common systemic form, is caused by patients’ unique misfolded light chains produced by a small B cell clone [2].

These light chains present mutations in the variable regions that cause low folding stability, high protein dynamics, and favor improper aggregation [3,4]. The production of misfolded proteins is followed by the formation of prefibrillar aggregates and finally fibrils, which may exert toxic effects on cells [5]. The gradual replacement of parenchymal tissue with amyloid deposits causes loss of function in target organs. In fact, it has been reported that the misfolded light chains may contribute to increase mitochondrial-related reactive oxygens species (ROS) production, impair intracellular calcium homeostasis, and induce cellular contractile dysfunction and morphological damage of mitochondria, with subsequent reduction of cell viability [6,7].

The mechanisms of amyloid deposition and target organ dysfunction are not yet fully understood. However, it has been hypothesized that the extracellular environment components could lead to amyloid proteolytic cleavage and binding to glycosaminoglycans and collagen, facilitating fibril accumulation [8].

AL amyloidosis is more commonly seen in association with a small-sized, otherwise indolent, bone marrow (BM)-residing plasma cell clone, but it can occasionally accompany a frankly malignant plasma cell clone fulfilling the features of multiple myeloma (MM), or be associated with IgM-producing B cell clones, such as Waldenström macroglobulinemia [9,10,11].

Growing experimental evidence indicates that plasma cell pathological behavior is strongly influenced by the BM microenvironment. One of the main cellular compartments constituting BM stroma is represented by mesenchymal stromal cells (MSCs), known to play a key role in the regulation of hematopoiesis and tumor growth [12,13,14].

It has been reported that a primary MSC defect could be responsible for disease pathological features in different hematological disorders [15,16], as demonstrated in BM-MSCs from myelofibrosis patients harboring genetic abnormalities and displaying altered functionality [17]. On the other hand, in MM, BM-MSCs display a lower proliferation rate and a reduced osteogenic capacity compared to normal MSCs, but these defects have been related to the reduced expression of several growth factor receptors, including EGFR, IGF1R, PDGFαR, and FGFR [18,19]. Additionally, it has been reported that myeloma cells in the BM niche take advantage of a pro-inflammatory microenvironment, especially characterized by the presence of high levels of interleukin-6 (IL-6) produced by BM stromal cells [20]. The release of growth factors, such as vascular endothelial growth factor (VEGF) and fibroblast growth factor (FGF), from MM cells would in turn stimulate IL-6 production by the BM-MSCs, determining a cytokine/growth factor amplification loop [21]. This linkage has been further confirmed by the detection of high IL-6 levels in BM-MSCs from newly diagnosed MM patients not receiving treatment [21].

In this work, BM-MSCs from AL patients were characterized following the criteria defined in the literature. We also investigated the key functional features by in vitro co-cultures with a plasma cell line or with circulating healthy donor B cells. The identification of specific defects in the main cellular population of BM stroma, known to support pathological plasma cell survival, will give the possibility to develop either more targeted therapies for AL or strategies to overcome the drug resistance characterizing the disease.

## 2. Materials and Methods

### 2.1. Enrollment of Patients and Healthy Donors

Diagnostic leftovers of BM aspirations from 57 consecutive patients with a final diagnosis of systemic AL amyloidosis evaluated at the Italian National Referral Center for Systemic Amyloidosis at the IRCCS Policlinico San Matteo Foundation, Pavia, Italy were included in the study. The diagnosis of AL amyloidosis and the definition of organ involvement was based on international consensus criteria [22]. Healthy donor MSCs (HD-MSCs) were obtained from residual BM cells harvested to be used in hematopoietic stem cell transplantation (HSCT). This source was available from 14 donors with a median age of 31.5 years (range 26–45). We are aware that the groups of patients and controls are not comparable for age, but we did not have the opportunity to enroll older healthy subjects undergoing this procedure.

### 2.2. AL-MSCs Isolation and Expansion

BM cells from both patients and HDs were collected as residual samples of the CD138 immuno-magnetic depletion performed for another ongoing research program. We conducted preliminary experiments assessing both intact and CD138 cell-depleted BMs, indicating that CD138 depletion did not alter either the AL- or HD-MSC expansion/differentiation capacity and their in vitro survival (data not shown). Therefore, we performed MSC isolation and expansion only from CD138neg BM-MNCs, following a standard in vitro culture procedure [23,24]. Briefly, MNCs were obtained from BM aspirates by density gradient separation and plated in polystyrene culture flasks (Corning Costar, Corning, NY, USA) at a density of 160,000/cm^2^ in DMEM + GlutaMAX (Gibco, Life Technologies, Milan, Italy) supplemented with 10% fetal calf serum (FCS) (Euroclone, Milan, Italy) (MSC-complete medium) and incubated at 37 °C, 5% CO_2_. This step was defined as P0 and culture medium was replaced twice a week. MSCs were harvested by Trypsin EDTA (Euroclone) at ≥80% confluence and re-plated for expansion at a density of 4000 cells/cm^2^.

### 2.3. Characterization of Ex Vivo Expanded AL-MSCs

AL-MSCs and HD-MSCs were characterized, as defined by Dominici et al. [25] by plastic adhesion, morphology, and clonogenic capacity (defined as CFU-F/10^6^ plated MNC at P0). Moreover, immune phenotype, osteogenic and adipogenic differentiation, and proliferation capacities were evaluated, as detailed below.

#### 2.3.1. Flow Cytometry

AL-MSC and HD-MSC surface antigens were evaluated at early and late passages by flow cytometry using CD73, CD90, CD105, and class I-HLA as positive markers and CD34, CD14, CD45, and CD31 (all antibodies from Beckman Coulter, IL, Milan, Italy) as negative markers [25]. Briefly, 1 × 10^5^ cells/tube were incubated at 4 °C with FITC or PE-conjugated monoclonal antibodies. In total, 10,000 events were acquired by FACS Navios flow-cytometer (Beckman Coulter). Analysis was performed by Navios software (Beckman Coulter).

#### 2.3.2. Differentiation Assay

The osteogenic and adipogenic differentiation capacity of AL-MSCs and HD-MSCs was evaluated at passage P2/P3 [24]. For osteogenic differentiation, the induction medium was αMEM, 10% FBS, 10^−7^ M dexamethasone, 50 mg/mL L-ascorbic acid, and 5 mM β-glycerol phosphate (all from Sigma Aldrich, St. Louis, MO, USA); for adipogenic differentiation, the induction medium was αMEM, 10% FBS, 10^−7^ M dexamethasone, 50 mg/mL L-ascorbic acid and 5 mM β-glycerol phosphate, 100 mg/mL insulin, 50 mM isobutyl methylxanthine (Sigma-Aldrich), and 0.5 mM indomethacin (MP Biomedica, Illkirch, France). In both protocols, differentiation was evaluated after 21 days. In vitro osteogenic differentiation was evidenced by phosphatase alkaline activity stained in blue/violet by BCIP/NBT and calcium deposition stained by Alizarin Red S (both from Sigma-Aldrich). In vitro adipogenic differentiation was evidenced by the appearance of fat droplets stained with Oil Red O (Bio Optica, Milan, Italy).

#### 2.3.3. Proliferative Capacity

Proliferative capacity was defined as cumulative population doubling (cPD) obtained by the sum of PD at different passages. PD was calculated by the formula:log_10_ (n. of harvested cells/n. of seeded cells)/log_10_(2)

#### 2.3.4. Senescence Assay

AL-MSCs and HD-MSCs senescence was assessed by a β-galactosidase (SA-β-gal) staining kit (Cell Signaling Technology, Danvers, MA, USA), according to the manufacturer’s instructions, and evaluated by direct-light microscopy.

#### 2.3.5. Gene Expression Profile

Total RNA from cellular pellets was extracted using the RNeasy Mini Kit (Qiagen, Hilden, Germany) according to the manufacturer’s instructions. RNA quality was quantified using a NanoDrop (Thermo Fisher Scientific, Waltham, MA, USA). A total of 1 μg of RNA per condition was reverse transcribed into cDNA (Reverse Trascritional M-MLV RT kit, Promega, Milan, Italy). cDNA was quantified spectrophotometrically using NanoDrop. Real-time PCR for *PDGαR*, *EGFR*, *FGFR*, and *IGF1R* genes was performed. Real-time PCR was performed on the Real-Time PCR instrument (AB 7500 Standard System). The glyceraldehyde 3 phosphate dehydrogenase (*GAPDH*) as the endogenous gene and predesigned PrimePCR SYBR Green assays were used (Bio-Rad, Hercules, CA, USA). Data analysis was done by 7500 fast Real-time PCR systems (Applied Biosystems, Waltham, MA, USA). Expression levels for each gene were calculated using the RQ method, normalizing the expression of gene of interest, with the expression of a reference gene (*GAPDH*) in the same sample.

### 2.4. ALMC-2 Cell Line

The ALMC-2 (RRID:CVCL_M526), a cell line established from a patient diagnosed with AL that exhibit a plasma cell phenotype, IL6-dependent growth, and free secretion of either λLC or IgG, was kindly provided by Dr. Jelinek [26]. ALMC-2 cells were cultured in IMDM + GlutaMAX supplemented with 10% heat-inactivated FBS and 1% pen/strep, 10 ng/mL IGF1, and 1 ng/mL IL-6 (all reagents from Gibco). Cells were seeded at a density of 2–5 × 10^5^ viable cells/mL and grown in a 37 °C incubator with 5% CO_2_. Cell cultures were split three times/week. Throughout the study, negativity to Mycoplasma spp. contamination was verified with the EZ-PCR Mycoplasma test Kit (Resnova, Rome, Italy) according to the manufacturer’s instructions.

### 2.5. Co-Culture of AL-MSCs, HD-MSCs, and ALMC-2

HD-MSCs (*n* = 8) and AL-MSCs (*n* = 6) were cultured in DMEM-complete medium and allowed to adhere to 96 well/plate, at 15,000, 1500, and 150 cells/well during overnight incubation at 37 °C, 5% CO_2_. The day after, 30,000 ALMC-2 cells were added, obtaining MSCs:ALMC-2 ratios of 1:2, 1:20, and 1:200, cultured in either IMDM-complete medium or DMEM-complete medium and incubated for 72 h at 37 °C, 5% CO_2_. Eighteen hours before harvest, 1 μCi/well ^3^H-thymidine (Perkin Elmer, Waltham, MA, USA) was added. Radioactivity was measured by gamma-counter (PerkinElmer). Experiments were performed in triplicate and results were expressed as count/minutes (cpm). The same co-culture experiments were performed to collect supernatants for the detection of λLC chains, IgG, and IL-6. After co-culture, ALMC-2 cells were evaluated for intracellular λLC chains.

### 2.6. Evaluation of λL Chain Levels in Culture Supernatants

λLC levels in culture supernatants produced by ALMC-2 in the presence or absence of AL, or HD-MSCs were detected by the automated nephelometric technique (Nephelometer BN ProSpec^®^ System, Siemens, Milan, Italy).

### 2.7. Quantification of IgG and IL-6 Levels in Culture Supernatants

IgG levels in culture supernatants were detected by ELISA. Briefly, microtiter plates (Greiner, Milan, Italy) were coated with polyclonal rabbit anti-human IgG (Dako, St. Clara, CA, USA) and incubated 3 h at 37 °C and then overnight at 4 °C. After washes, culture supernatants were incubated for 2 h at 37 °C and horse-radish peroxidase (HRP)-conjugated rabbit anti human IgG (Dako) was added. After 2 h of incubation, substrate solution (o-Phenylenediamine dihydrochloride) (Sigma Aldrich) was added. Colorimetric reaction was read at 492 nm (Sunrise, Tecan, Switzerland). Data were calculated by Magellan Software (Tecan). IgG levels were expressed as µg/mL.

IL-6 levels were detected by ELISA using Duo set antibody pairs (R&D System, Minneapolis, MN, USA) according to the manufacturer’s instructions. Results were expressed as pg/mL.

### 2.8. Intracellular λLight Chains

After co-culture of ALMC-2 cells with AL- and HD-MSCs from HD, the intracellular expression of λLC was evaluated by flow cytometry. Briefly, 2 × 10^5^ ALMC-2 cells were washed in PBS at 2000 rpm for 5 min. ALMC-2 cells were fixed (Fix and Perm, Nordic MUbio, Susteren, The Netherlands) and during the permeabilization step, the Cy-IgL monoclonal antibody was added (3 μL) at RT for 15 min. ALMC-2 cells were washed and acquired. Results expressed as mean fluorescence intensity (MFI) were shown as ratio of the MFI of ALMC-2 cells co-cultured with MSCs/MFI of ALMC-2.

### 2.9. Co-Culture of AL-MSCs, HD-MSCs, and Peripheral Blood Mononuclear Cells (PBMCs)

In order to evaluate the effect of MSCs on IgG secretion by circulating B cells, PBMCs were isolated by density gradient from one HD and cultured in IMDM (Gibco) supplemented with 1 mg/mL bovine insulin (Sigma Aldrich), 0.1% bovine albumin (Sigma Aldrich), Hepes buffer 0.01 M (Sigma Aldrich), 1 mM glutamine (Gibco), 5% FCS (Euroclone), and 1 g/mL ethanolamine (Sigma Aldrich) as previously described [27]. Briefly, 1 × 10^5^ PBMCs/well were incubated in the presence or absence of AL and HD-MSCs at different MSCs:PBMCs ratios (1:2, 1:20, and 1:200). After 13 days at 37 °C, ELISA as described above collected 5% CO_2_ supernatants for IgG quantification. The results were compared to those obtained in the same settings using HD-MSCs.

### 2.10. Statistical Analysis

Multilevel population average generalized estimating equation (GEE) models with first-order autocorrelation were fitted to take into account the clustered nature of the data. The cellular proliferation rate, IgG, λLC, and IL-6 were dependent variables while MSC source (AL patients and HD) and MSC concentration ratios were predictors.

Senescence was evaluated comparing the senescence passage of AL-MSCs and HD-MSCs by Kruskal–Wallis non parametric analysis of variance followed by 2 × 2 post hoc comparisons. After fitting a repeated measure model, the proliferative capacity expressed as cPD was evaluated by comparing the number of AL-MSCs and HD-MSCs at different passages by Kruskal–Wallis non parametric analysis of variance followed by 2 × 2 post hoc comparisons. The expression of genes encoding for the growth factor receptors, *EGFR, PDGFαR, FGFR*, and *IGF1R* was evaluated by fitting GEE models to take into account the clustered nature of the data, where the expression of genes is dependent variables and group (AL patients and HD) is an independent variable.

## 3. Results

Fifty-seven consecutive patients with a final diagnosis of systemic AL amyloidosis were enrolled in the study. Patient characteristics are summarized in Table 1. Results were compared to those obtained from 14 HD-MSCs.

### 3.1. Characterization of Ex Vivo Expanded AL-MSCs

AL- and HD-MSCs were plastic adherent, showed the typical spindle shape morphology (Figure 1A), and were able to differentiate into osteoblasts and adipocytes as demonstrated by the detection of phosphatase alkaline activity and calcium depositions (Figure 1B), and by the morphologic appearance of lipid droplets (Figure 1C).

As HD-MSCs, AL-MSCs expressed ≥95% of CD90, CD73, CD105, and HLA-I surface antigens, and ≤5% of CD34, CD45, CD14, and CD31 molecules (Figure 2).

BM clonogenic capacity, expressed as the number of CFU-F/10^6^ plated cells, was evaluated at P0, after 7 days of culture. No statistical difference in CFU-F was observed between AL- and HD-MSCs (7.16 ± 7.76 and 6.84 ± 6.65 mean ± SD, respectively). On the contrary, AL-MSCs displayed a significantly reduced proliferative capacity from passage P3 to P7 compared to HD-MSCs. In particular, the MSC proliferative capacity, expressed as cPD, was evaluated in AL patients and HDs from P1 to the passage of senescence (Figure 3A). Since senescence was reached at different passages in AL and HD-MSCs, the percentage of subjects at each passage is reported in Figure 3B.

As shown in Figure 4, AL-MSCs entered senescence at a statistically significant (*p* < 0.0001) earlier passage (median P4, range P1-P10) than HD-MSCs (median P12, range P4-P21).

#### Expression of Growth Factor Receptors in MSCs

In order to explain the reduced proliferative capacity of AL-MSCs, we analyzed the expression of genes encoding for growth factor receptors, such as EGFR, PDGFαR, FGFR, and IGF1R, that have been reported to be relevant in MM [18]. In AL-MSCs, we observed a significant downregulation (*p* = 0.036) of EGFR gene expression compared to HD-MSCs, while FGFR, IGF1R, and PDGαR gene expression was comparable in AL-MSCs and HD-MSCs (Figure 5).

### 3.2. Modulatory Effect of MSCs Co-Cultured with ALMC-2 Cells

AL- or HD-MSCs were co-cultured at different ratios with ALMC-2 cells in the presence of MSC-specific culture medium (DMEM with 10% FCS indicated as DMEM complete medium) or ALMC-2-specific culture medium (IMDM with IL-6 and IGF1, indicated as IMDM complete medium) to evaluate the MSC modulatory effect on ALMC-2 cell proliferation and *λ*LC and IgG secretion.

MSC/ALMC-2 cell co-culture experiments performed in DMEM complete medium showed that both AL- and HD-MSCs significantly increased ALMC-2 cell proliferation at 1:20 and 1:200 ratios (*p* = 0.009; *p* < 0.0001, respectively, and for all), while *λ*LC secretion was reduced in a dose-dependent manner at a 1:2 ratio in the presence of either AL- or HD-MSCs, and at a 1:20 ratio in presence of AL–MSCs (*p* < 0.0001). IgG secretion was not affected by the presence of either AL- or HD-MSCs (Figure 6A–C, respectively).

On the contrary, when MSC/ALMC-2 cell co-culture experiments were performed in IMDM complete medium, we observed that both patients and HD-MSCs significantly inhibited ALMC-2 cell proliferation and *λ*LC and IgG secretion at 1:2 and 1:20 ratios (*p* < 0.0001, for all) (Figure 6D–F, respectively).

A limited number of co-culture experiments of ALMC-2 cells and AL- (*n* = 3) or HD-MSCs (*n* = 4) was performed to assess the intracellular production of *λ*LC in ALMC-2 cells. No significant difference was observed in the presence of AL- or HD-MSCs (data not shown).

To assess whether MSCs were able to influence the nature of the culture medium by secreting inflammatory factors, IL-6 levels were measured in co-culture supernatants of AL- or HD-MSCs and ALMC-2 cells cultured in DMEM complete medium. IL-6 levels were comparable in the co-culture supernatants of the two groups. In addition, IL-6 levels were comparable in culture supernatants collected from AL- or HD-MSCs alone at the concentration of 15,000 MSCs/well (number of MSCs in the 1:2 ratio), 1500 MSCs/well (number of MSCs in the 1:20 ratio), and 150 MSCs/well (number of MSCs in the 1:200 ratio). As expected, significant differences were observed among the different cultured MSC concentrations (Figure 7). No IL-6 was detected in the supernatant of ALMC-2 cells alone.

### 3.3. Modulatory Effect of MSCs Co-Cultured with HD-PBMCs

In order to evaluate the effect of MSCs on IgG secretion by circulating B cells, we co-cultured HD-PBMCs with AL-MSCs or HD-MSCs. IgG production by resting PBMCs was significantly enhanced by both AL- and HD-MSCs at 1:2 and 1:20 ratios (*p* = 0.004 and *p* < 0.0001, respectively, and for all) compared to PBMCs alone. No difference was observed between the groups (Figure 8).

## 4. Discussion

Alterations within the BM microenvironment may either lead to or favor the pathogenesis of different hematologic disorders. MSCs represent one of the key components of the BM niche. They are multipotent cells with well-documented immune regulatory functions on both innate and adaptive immunity [28,29]. The capacity of MSCs to inhibit T cell responses is well established, while different results have been reported regarding their ability to regulate B lymphocyte functions. Some studies documented their inhibitory effect on B cell proliferation, differentiation, and immunoglobulin secretion, while others demonstrated stimulatory effects [30]. It is known that the BM niche plays a crucial role in supporting plasma cell growth, proliferation, and survival both in physiological and pathological conditions [15,17,18]. The BM compartment has been the subject of several studies in the context of MM [31,32], while a detailed characterization of the BM niche in the context of AL amyloidosis is missing.

In this study, we characterized MSCs isolated and expanded from BM of AL patients at the disease onset, not yet receiving therapy. Our results showed that AL- were comparable to HD-MSCs for morphology and immune phenotype. In addition, differently from what has been reported in other clonal plasma-cell disorders, the capacity to differentiate in adipocytes or osteocytes was comparable to that of HD-MSCs [31].

Significant differences in terms of proliferation were reported in MSCs expanded from MM patients with respect to the normal counterpart [18,32]. Similarly, we observed that AL-MSCs showed a lower proliferation rate than HD-MSCs [32]. To this regard, it has been demonstrated in MM that the in vitro MSC proliferation defects could be due to downregulated expression of genes related to proliferation and senescence, such as those encoding for *EGFR*, *IGF1R*, *FGFR*, and *PDGF**αR* [18,33,34,35,36,37]. In AL-MSCs, we observed the significant downregulation of the *EGFR* gene, whose activation in human HD-MSCs has been reported to either increase cell proliferation or prevent adipogenic, osteogenic, and chondrogenic differentiation [34,35]. No alteration in the gene expression of *IGF1R*, *FGFR*, and *PDGFαR* characterized AL-MSCs; actually, these genes have been described to be involved also in the MSC differentiation process that is not impaired in AL-MSCs [36,37,38].

The modulatory function of AL-MSCs was assessed by co-culture with ALMC-2 cells. This cell line is recognized as a useful tool to study the biology of plasma cells secreting amyloidogenic λLC [26]. The AL-MSC modulatory effect on the ALMC-2 cell line was comparable to that of HD-MSCs. In particular, the co-cultures in the presence of DMEM complete medium resulted in an increased proliferation of ALMC-2 cells, decreased λLC secretion, and no effect on IgG secretion from ALMC-2 cells. These results confirm those reported by Gupta and coworkers, who co-cultured human MM cell lines in RPMI medium with BM stromal cells [39]. Conversely, the presence of IMDM complete medium with the addition of growth factors in co-cultures induced an inhibitory effect on both ALMC-2 cell growth and IgG/λLC secretion. It has been reported that the immune function of MSCs can be modulated by inflammatory priming in the surrounding environment [40], which represents a critical aspect for MSC immunosuppressive capacities [41,42,43], as previously reported in MM patients [23]. Our results confirm that the presence of inflammatory growth factors in ALMC-2 culture medium (i.e., IL-6 and IGF1) reverses the MSC regulatory activity. This effect is comparable in AL- and HD-MSCs.

To our knowledge, no data have been reported regarding the inflammatory/anti-inflammatory nature of the AL BM environment. Recent studies observed that IL-6 plays a central role in the creation of an ideal microenvironment for oncogenesis and metastasis in MM, due to its capacity to regulate bone homing, disease progression, and drug resistance [44]. Our study indicates that the IL-6 amounts secreted by AL-MSCs in culture supernatants in the presence/absence of ALMC-2 were comparable to those released by the HD-MSCs in the same culture conditions. This finding could be important to understand the quality of the BM microenvironment in AL pathogenesis, indicating that, differently from what was shown in MM [32,39,41], MSCs do not have a prominent role in shaping IL-6-mediated inflammatory conditions in BM of patients with AL amyloidosis.

Further characterizing AL-MSC capacity, we assessed their ability to stimulate IgG secretion by circulating B cells. Since the experiments were not performed with purified B cells, the data indicate that MSCs were able to support B cells probably cooperating with T cells according to published data obtained using HD-MSCs [44,45,46].

A decrease of proliferation, immunomodulation, and differentiation capacity, associated with an increase of IL-6 secretion, has been extensively described in MSCs from old healthy subjects [47,48]. We are aware that the age of our AL patients is significantly higher than that of HDs, and this represents a limitation of this study; nevertheless, in our hands, AL-MSCs showed only a decreased proliferation capacity leading to an earlier senescence, while immunosuppressive abilities, differentiation capacities, and IL-6 secretion are conserved functions, indicating no relation with age, and possibly representing a peculiar aspect of AL. Follow-up studies may help to understand if and how the disease course and severity may alter MSC functions changing this scenario.

## 5. Conclusions

This is the first study providing a comprehensive characterization of BM-MSCs in AL amyloidosis at the disease onset. Our data indicate that AL-MSCs undergo premature senescence, while maintaining a normal morphology, immune phenotype, and differentiation capacities. In addition, the ability to modulate the proliferation and IgG/λ chain secretion of ALMC-2 cells and the IgG production by circulating B cells is comparable to those of HD-MSCs. The lack of production of abnormal amounts of IL-6 by AL-MSCs suggests a different role for these cells in shaping the BM microenvironment with respect to what has been described in MM. This study could pave the way for a deeper identification of BM microenvironment components in AL, as a target for the prevention/treatment of the disease in personalized therapies.

## Figures and Tables

**Figure 1 biomedicines-09-01523-f001:**
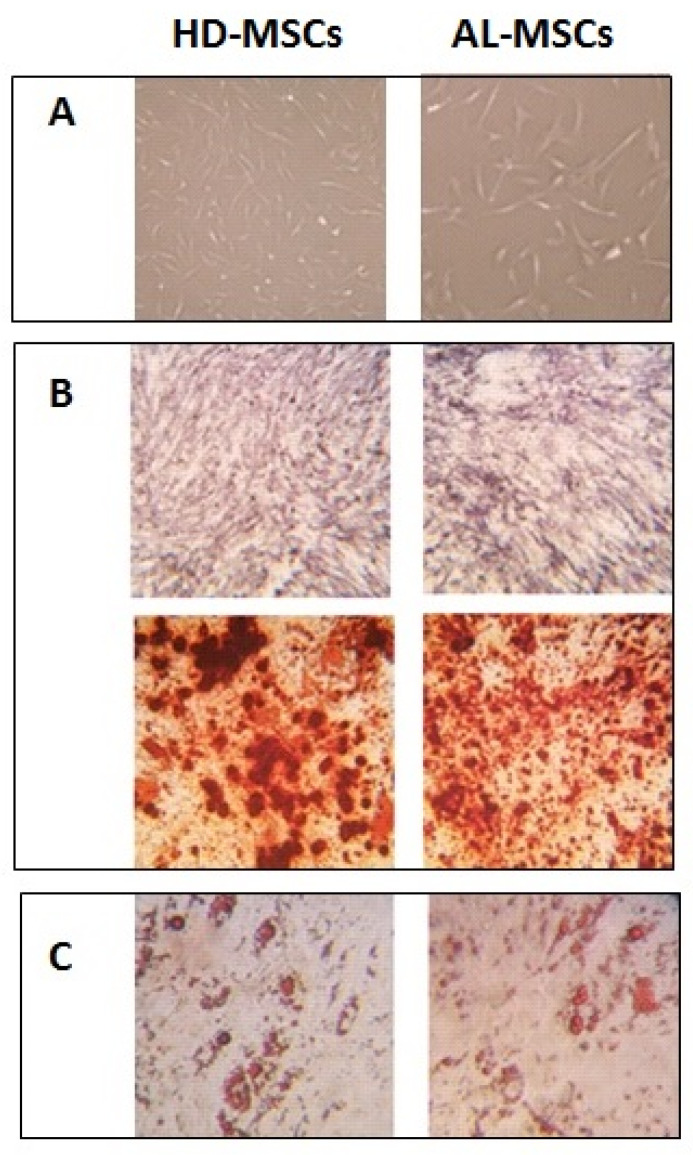
Morphology and differentiation capacity of AL-compared to HD-MSCs. (**A**) Typical spindle shape morphology of plastic adherent MSCs (4×). (**B**) In vitro-induced osteogenic differentiation demonstrated by the detection of phosphatase alkaline activity stained in blue/violet by BCIP/NBT and calcium deposition stained by Alizarin Red S (4×). (**C**) In vitro-induced adipogenic differentiation demonstrated by the appearance of lipid droplets stained by Oil red O (4×).

**Figure 2 biomedicines-09-01523-f002:**
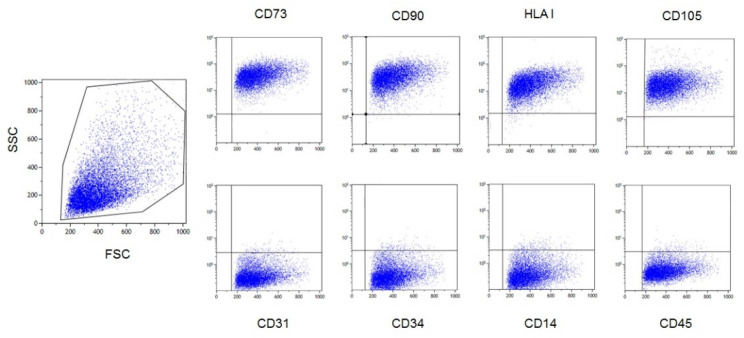
Flow cytometry analysis of one representative MSC lot. Positive markers (≥95% of the gated cell populations defined by physical parameters) and negative markers (≤5%) are reported in the upper and lower panels, respectively.

**Figure 3 biomedicines-09-01523-f003:**
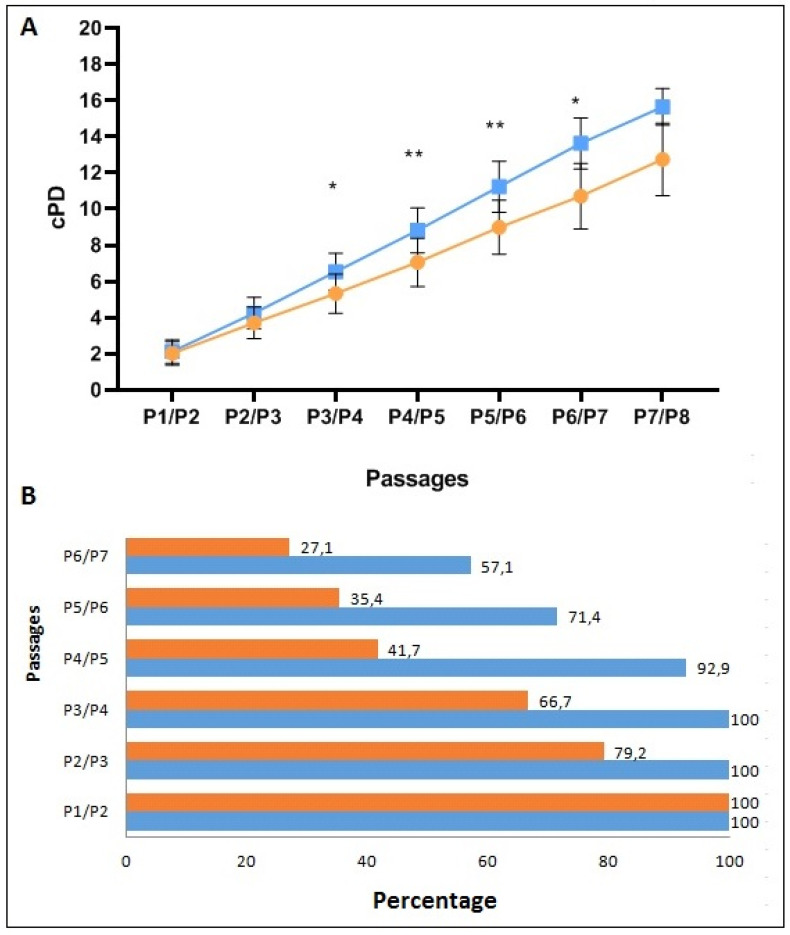
Proliferation capacity of AL- and HD-MSCs at different culture passages. (**A**) Results, expressed as cPD calculated from P1/P2 to P7/P8 in AL-MSCs (orange line) and in HD-MSCs (blue line), are shown as mean ± SD (* *p* ≤ 0.05; ** *p* ≤ 0.005). (**B**) The percentage of subjects at each culture passage is shown at the right end of the bars. AL-MSCs (orange bar), HD-MSCs (blue bar).

**Figure 4 biomedicines-09-01523-f004:**
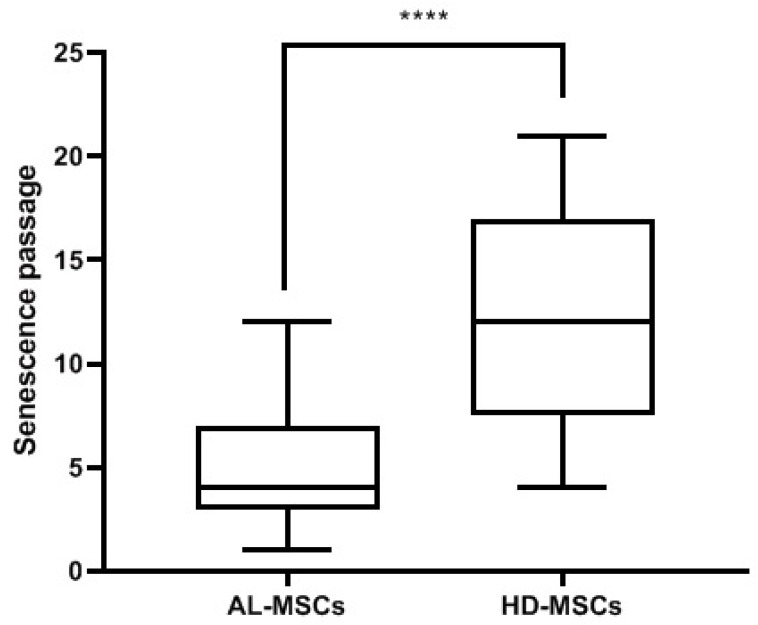
Senescence passages of AL- and HD-MSCs. Medians of the results [25–75%, Min-Max] are reported. **** *p* < 0.0001.

**Figure 5 biomedicines-09-01523-f005:**
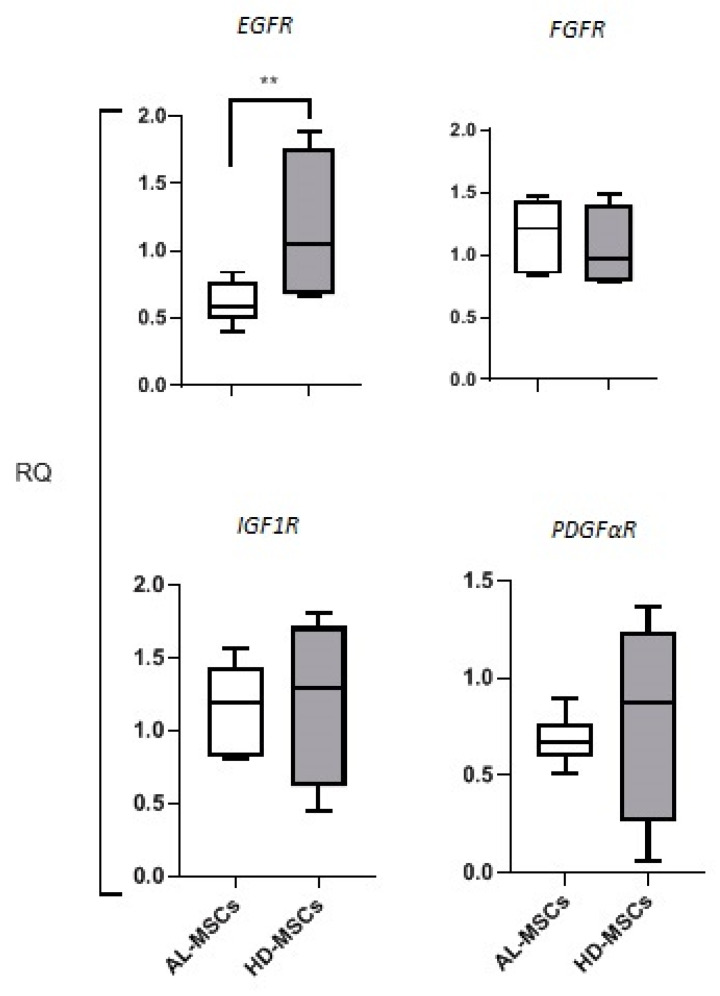
Relative expression pattern of genes encoding for growth factor receptors. The evaluation of transcript levels for genes encoding EGFR, FGFR, IGF1R, and PDGαR was performed in AL- and HD-MSCs by semiquantitative RT-PCR. The expression level for each gene was calculated using the RQ method, normalizing the expression of the gene of interest with the expression of a reference gene (GAPDH) in the same sample. Results are shown as median [25–75%, Min-Max]. ** *p* = 0.036.

**Figure 6 biomedicines-09-01523-f006:**
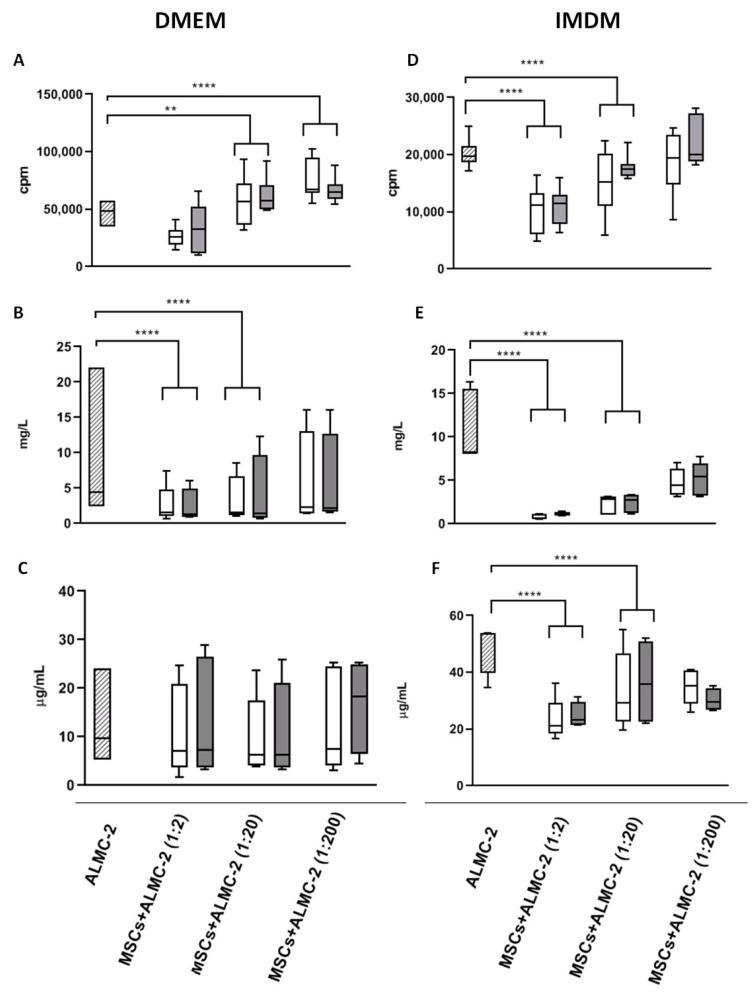
MSC/ALMC-2 co-culture experiments. AL-MSCs (*n* = 8, white bar) and HD-MSCs (*n* = 7, grey bar) were co-cultured at different ratios with the ALMC-2 cell line (dashed bar). Cultures were performed in MSC (DMEM) (left panels) or ALMC-2 (IMDM) (right panels) complete media. ALMC-2 cell proliferation, expressed as cpm for ^3^H Thymidine incorporation (**A**,**D**). ALMC-2 *λ*LC secretion (**B**,**E**). ALMC-2 IgG secretion (**C**,**F**). Results are shown as median [25–75%, Min-Max]. ** *p* = 0.009; **** *p*< 0.0001.

**Figure 7 biomedicines-09-01523-f007:**
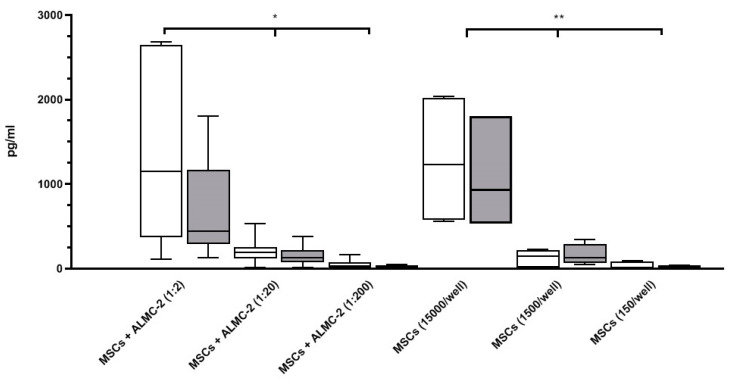
IL-6 production in AL- and HD-MSC co-culture supernatants. Evaluation of IL-6 production by AL-MSCs (white bar) and HD-MSCs (grey bar) after co-culture with ALMC-2 cells at different ratios in DMEM complete medium. IL-6 was also quantified in the supernatants of AL- and HD-MSCs alone at the same number of MSCs for the different ratios. Results, expressed as pg/mL, are shown as median [25–75%, Min-Max]. * *p* = 0.01 for all, ** *p* = 0.003 for all.

**Figure 8 biomedicines-09-01523-f008:**
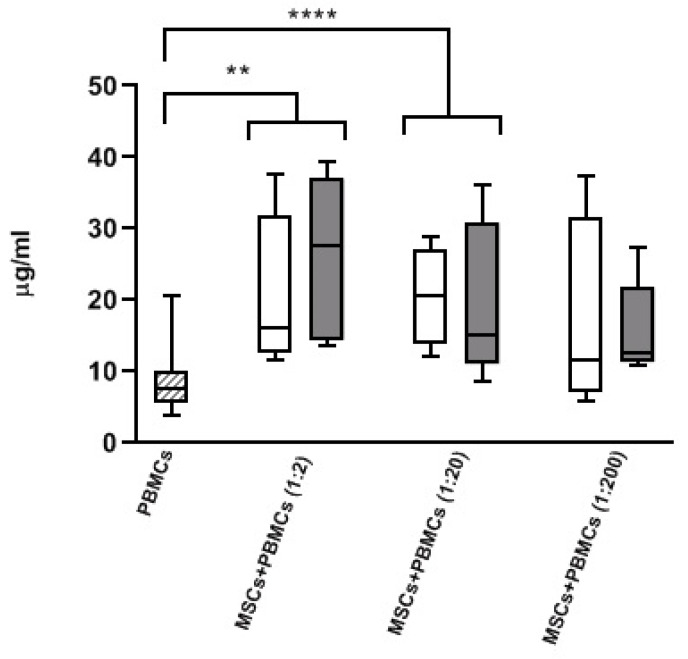
MSCs co-cultured with resting PBMCs. IgG secretion by PBMCs of one HD was quantified after co-culture with AL- (white bar) and HD-MSCs (grey bar) at different ratios. Results were compared to PBMCs alone (dashed bar)**.** Results, expressed as µg/mL, are shown as median [25–75%, Min-Max]. ** *p* = 0.004, **** *p* < 0.0001.

**Table 1 biomedicines-09-01523-t001:** Clinical characteristics of AL patients included in this study.

**Subjects**		n (%)	
Patients		57	
Male sex		33 (57.9)	
		**Median (IQR)**	
Age, yrs		67 (60–73)	
% BMPC *		9 (6–13)	
dFLC, mg/L **		175.3 (79.1–444.8)	
**Organ**		n (%)	
Heart		43 (75.4)	
Kidney		27 (47.4)	
Soft tissues		13 (22.8)	
Liver		3 (5.3)	
**M protein**			
Type	(n)	Type	(n)
IgAκ	1	IgAλ	7
IgGκ	4	IgGλ	12
IgMκ	0	IgMλ	3
κ	9	λ	21

* % BMPC: percentage of BM plasma cell infiltrate, ** dFLC: difference between involved and uninvolved serum-free light-chain levels.

## Data Availability

Not applicable.

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
