# Peer review of "Bone Marrow Microenvironment in Light-Chain Amyloidosis: In Vitro Expansion and Characterization of Mesenchymal Stromal Cells"

_biomedicines, 2021, doi:10.3390/biomedicines9111523_

Round 1

Reviewer 1 Report

This is a good work with high degree of novelty. The manuscript is well written with adequate description of the material and methods section, results and discussion. However there are few typos throughout the manuscript which require correction. Furthermore, I would recommend using PDGFRA, EGFR, FGFR1, IGF1R forms in the manuscript with names in italics for gene description or expression, and regular form for protein level based expression. I would not recommend using e.g. the EGF-r form. Besides I have no other comments.

Author Response

We thank the reviewer for the comments. As suggested, we corrected  the typos throughout the manuscript. The genes’  acronyms have been modified and are now written in Italics. The r for receptor is now written in capital letter.

Reviewer 2 Report

I consider that the idea of this study, to evaluate the BM-MSCs from AL patients, is very interesting and with important clinical implications. I consider that the findings are interesting and that the results obtained can make significant contributions to further large studies. The article is well-written and comprehensive, with clear and legible tables and it may be accepted for publication after some minor revisions. I suggest highlighting the gaps of the study and also research priorities in this field. I also consider that the manuscript would benefit enormously from English editing because there are still some leaks.

Author Response

We thank the reviewer for the comments. As suggested, we improved the description of the research design at the end of the introduction and material and method section. Moreover, we highlighted both the gaps (material and methods section and at the end of the discussion) and the research priorities of the study (end of the discussion). The English editing was done and the leaks have been corrected throughout the manuscript.

Round 2

Reviewer 2 Report

The authors have significantly improved the article and I consider that now it can be published in its current form.